# Using an integrated knowledge translation approach to inform a pilot feasibility randomized controlled trial on peer support for individuals with traumatic brain injury: A qualitative descriptive study

**Stephanie K. C. Lau**[1]*, **Dorothy Luong**[2], **Shane N. Sweet**[3,4], **Mark Bayley**[2,5], **Ben B. Levy**[6], **Monika Kastner**[7], **Michelle L. A. Nelson**[5,8], **Nancy M. Salbach**[2,9,10], **Susan B. Jaglal**[2,5,9,10], **John Shepherd**[10], **Ruth Wilcock**[11], **Carla Thoms**[11], **Sarah E. P. Munce**[2]

1 Faculty of Medicine, University of Ottawa, Ottawa, Ontario, Canada, 2 Toronto Rehabilitation Institute, University Health Network, Toronto, Ontario, Canada, 3 Department of Kinesiology and Physical Education, McGill University, Montreal, Quebec, Canada, 4 Center for Interdisciplinary Research in Rehabilitation of Greater Montreal, Montreal, Quebec, Canada, 5 Institute of Health Policy, Management, and Evaluation, University of Toronto, Toronto, Ontario, Canada, 6 Temerty Faculty of Medicine, University of Toronto, Toronto, Ontario, Canada, 7 North York General Hospital, Toronto, Ontario, Canada, 8 Bridgepoint Collaboratory, Lunenfeld-Tanenbaum Research Institute, Toronto, Ontario, Canada, 9 Department of Physical Therapy, University of Toronto, Toronto, Ontario, Canada, 10 Rehabilitation Sciences Institute, University of Toronto, Toronto, Ontario, Canada, 11 Ontario Brain Injury Association, St. Catharines, Ontario, Canada

* SLau065@uottawa.ca

## Abstract

### Introduction

Traumatic brain injury (TBI) is estimated to affect 10 million people annually, making it a leading cause of morbidity and mortality worldwide. One cost-effective intervention that has been shown to minimize some of the negative sequelae after TBI is peer support. However, the evidence supporting the benefits of peer support for individuals with TBI is sparse and of low quality. Integrated knowledge translation (iKT) may be one approach to optimizing the evaluation of peer support programs among individuals with TBI. Therefore, the objectives are: (1) To understand key informants' perspectives of the barriers and facilitators of participating in peer support research and programs among individuals with TBI; (2) to understand key informants' perspectives on the perceived impacts of peer support programs on individuals with TBI; and, (3) to demonstrate how an iKT approach can inform the development and implementation of a pilot feasibility randomized controlled trial (RCT).

### Methods

A qualitative descriptive approach using one-on-one semi-structured interviews was used. Purposive sampling of 22 key informants included 8 peer support mentors, 4 individuals with TBI who received peer support, 3 caregivers of individuals with TBI, 4 peer support program staff, and 3 academics in peer support and/or TBI.

**Data Availability Statement:** All relevant data are within the manuscript and its Supporting information files. Full transcripts cannot be shared publicly as the participants did not consent to such. However, researchers can request for access to these full transcripts via the University Health Network's Research Ethics Board (contact via reb@uhnresearch.ca).

**Funding:** This work was supported by an Ontario Neurotrauma Foundation Grant (grant number 2017- 33 ABI-OBIA-1036), received by SM. The funders had no role in study design, data collection and analysis, decision to publish, or preparation of the manuscript.

**Competing interests:** The authors have declared that no competing interests exist.

## Results

There were five main themes related to the barriers and facilitators to participating in peer support research and programs: knowledge, awareness, and communication; logistics of participating; readiness and motivation to participate; need for clear expectations; and matching. There were three main themes related to the perceived impact of peer support: acceptance, community, social experiences; vicarious experience/learning through others: shared experiences, role-modelling, encouragement; and "I feel better." Discussions with our Research Partner led to several significant adaptations to our trial protocol, including removing the twice/week intervention arm, shortening of the length of trial, and changing the measure for the community integration outcome.

## Discussion/Conclusion

This is the first study to use an iKT approach to inform a trial protocol and the first to assess the barriers and facilitators to participating in peer support research.

## Introduction

Traumatic brain injury (TBI) is estimated to affect 10 million people annually, making it a leading cause of morbidity and mortality worldwide [1, 2]. In addition to the physical and cognitive sequelae, individuals with TBI often experience significant psychological distress as a result of the abrupt and often dramatic alterations to their day-to-day lives [3]. The inability to cope with these sudden changes may lead to depression, anxiety, and reduced quality of life [3]. Individuals with TBI often feel socially isolated and experience difficulty reintegrating into the community [3, 4]. Approximately 43% of individuals who suffer from a TBI develop long-term disability, causing a significant financial strain on the healthcare system [5]. However, clinical studies have shown that while medical interventions reduce the mortality from TBI, there has been limited demonstrated improvement on the functional outcome of survivors of moderate to severe TBI [6].

One cost-effective intervention that has been shown to minimize social isolation and potentially promote community reintegration after TBI is peer support [7]. Peer support is defined as the provision of knowledge and support by a person with a similar health condition and experience as the person they are assisting [7–9]. Peer support has been shown to be an effective service for individuals with TBI, including improvements in mood, perceived social support, behavioural control, empowerment, coping, and quality of life [10]. However, the impact of peer support on community reintegration [10–12], and the 'active ingredients,' or factors that lead to the success of peer support for individuals with brain injury remain unclear [10]. Furthermore, the evidence supporting the benefits of peer support for individuals with TBI is sparse and of low quality. For example, systematic reviews on the impact of peer support for individuals with TBI [10–12] have identified only six studies, of which only two of these studies were of high-quality design [13, 14]. Thus, more randomized controlled trials (RCTs) in this area are needed, but they are often difficult to implement in rehabilitation contexts due to the complex nature of the interventions and inability for double-blinding [15].

Integrated knowledge translation (iKT) may be one approach to optimizing RCT protocols to evaluate peer support among individuals with TBI. iKT is defined as an active and dynamic collaboration between researchers and key informants (knowledge users, or those with the

authority to enact change) in the synthesis, dissemination, and application of knowledge [16]. The rapid uptake of iKT in health service research is in response to the growing gap between research production and research utilization in healthcare [17–20]. The implementation of iKT during the research process can expedite the application of research outcomes into practice or policy as the use of an iKT approach increases the *accessibility* (knowledge users' ability to understand and apply the research), *relevance* (applicability of the research questions to current concerns), and *endurance* (sustainability of changes associated with the research findings because of the long-term partnerships that are formed from these collaborations) of research outcomes [17, 18, 21, 22]. Three recent scoping reviews on the use of iKT in research have demonstrated the importance of iKT in generating sustainable and relatable interventions [17, 18, 22]. In all three reviews, the majority of studies included had positive outcomes associated with iKT use, such as an increase in the relevance and quality of research findings; an increase in the sharing and uptake of findings; a formation of long-term partnerships that facilitate future collaborations; an exposure to different perspectives on the scope of research; and an increase of stakeholders' insight into their own needs, available community resources, and ability to advocate [17, 18, 22]. However, while positive outcomes with iKT use has been documented, the fidelity to the iKT approach (i.e., true engagement of key informants) and its potential, resulting impact on the research *process* are less clear because they are seldom described in detail in the literature [23, 24]. Specifically, studies reporting the use of iKT often lack information on key informants' involvement and influence in the data recruitment and collection processes [17, 25, 26]. For example, in the review by Gagliardi and colleagues [17], of the thirteen studies included, only two had mentioned whether stakeholders were involved in the data recruitment and collection step, and only five had mentioned whether there was stakeholder involvement in the interpretation of research findings. With respect to the studies describing involvement in recruitment and data collection, neither of them described how the stakeholders' involvement influenced the process [25, 26]. Moreover, one study had surveyed the stakeholders and researchers regarding their involvement but none of the stakeholders (0/9) and only five of the 26 researchers agreed or strongly-agreed that the data collection phase was a joint effort [26]. The review by Camden and colleagues [22] postulated that the omission of stakeholder engagement information in the identified studies might be because stakeholders were simply *informed* throughout the research process, without an actual opportunity to engage and influence the process.

Thus, the objectives of the current study are to: (1) understand key informants' perspectives of the barriers and facilitators of participating in peer support research and programs among individuals with TBI; (2) understand key informants' perspectives on the perceived impacts of peer support programs on individuals with TBI; and, (3) demonstrate how an iKT approach can inform the development and implementation of a pilot feasibility RCT.

The third objective will be achieved by using the findings from the first and second objectives to inform a pilot feasibility RCT in collaboration with our Research Partner, the Ontario Brain Injury Association (OBIA). As described in the published protocol [27], OBIA's Peer Support Program uses a one-to-one model of peer support and the recipients of peer support are referred to as 'partners.'

## Methods

### Study design

The protocol for this study has been previously published [27]; herein we provided a condensed version of the methods. Ethics approval was obtained from the University Health

Network Research Ethics Board. A qualitative descriptive approach, grounded in a pragmatic paradigm, was adopted [28–31].

## Participants and recruitment

A purposive sampling of a variety of key informant groups was used, including individuals with moderate-to-severe TBI (partners), caregivers, the OBIA Peer Support Program mentors, OBIA Peer Support Program staff members, health services and knowledge translation researchers with expertise in TBI and methodologists with expertise in clinical trials. Eligible partner participants, including caregiver partners, included community-based (i.e., no longer participating in a comprehensive rehabilitation program) individuals who (1) had a moderate-to-severe TBI (Glasgow Coma Scale ≤12) [32] or had cared for (i.e., unpaid) an individual with moderate-to-severe TBI for at least 1 year, (2) were at least 18 years of age, (3) had participated in the OBIA Peer Support Program, (4) were fluent in English and (5) were able to provide informed written consent or had a proxy to provide informed written consent. To be eligible, mentor participants must have completed at least one partnership with the OBIA Peer Support Program. Exclusion criteria included individuals who were medically unstable or had active suicidal ideation. The OBIA Peer Support Program database was used to recruit the partners, caregivers, and mentors. The OBIA staff members were recruited using the contacts of the investigators, and the researchers and methodologists were recruited through online searches. Emails were used for recruitment. Recruitment ceased at point of data saturation.

## Data collection

Participants took part in a one-on-one, semi-structured telephone interview lasting approximately 45–60 minutes. The interview guide that was used is included in the S1 File and it was pilot tested with various members of the research team with experience in qualitative methods, an individual with moderate and severe TBI, as well as an existing mentor. The interviews focused primarily on the barriers and facilitators to participating in peer support programs and research as well as the perceived impacts of participating in peer support. The principal investigator (SM) and research coordinator (DL) conducted a subset of interviews together, and DL conducted the remaining interviews alone. No one else was present for the interviews. Some of the researcher and methodologist participants were within the professional network of SM but no other prior relationships were established. As a part of the informed consent process, participants learned the objectives of the study, and the name and role of the interviewer. SM is a female Scientist with a PhD in Health Services Research, expertise in knowledge translation, and 12 years of experience conducting qualitative research. DL is a female research associate with a MSc in Rehabilitation Science and 12 years of experience conducting qualitative studies. There were no repeat interviews. All interviews were digitally recorded and transcribed verbatim. The transcripts were not returned to participants for feedback.

## Data analysis

Both inductive and deductive analyses were used [33, 34] in NVivo V. 11. The principal investigator (SM) and research coordinator (DL) independently coded a subset of transcripts, DL then developed a coding framework and applied it to the remaining transcripts. Following this, the codes were clustered into groups or categories and the predominant themes were identified. SM and DL explored various thematic maps until consensus was reached and the theme labels were agreed on.

Through bimonthly telephone meetings, the preliminary analyses, key findings, and final analyses were shared and reviewed with OBIA and their feedback was used to adapt the pilot feasibility trial protocol prior to its implementation.

## Results

A total of 22 key informant interviews were conducted: 8 mentors of the OBIA Peer Support Program, 4 partners of the OBIA Peer Support Program, 3 caregivers, 4 individuals with experience working in peer support programs, and 3 clinicians and/or researchers with expertise in peer support and/or TBI. There were no refusals to participate or dropouts.

### Barriers and facilitators to participating in peer support research and programs

Five barriers and facilitators to participating in peer support research and programs were identified. The first two themes were deductive (i.e., derived from probes from the interview guide) and the remaining three were inductive. Across these themes, the overarching considerations of memory, energy, and need for flexibility among individuals with TBI was noted. Table 1 presents representative quotes for each theme.

**Knowledge, awareness, and communication.** A lack of knowledge or awareness of peer support programs was a barrier to participating in such programs. Strategies, including having healthcare professionals provide information about the program and receiving information about the program earlier in recovery, were suggested to increase general, and earlier, participation in peer support programs. Similar strategies were suggested to facilitate research recruitment. Clear communication (e.g., using plain language) from community partners, local peer support coordinators, and researchers were noted as important strategies to optimize recruitment and participation in research.

**Logistics of participating.** Logistical concerns such as accessibility, convenience, and no/minimal monetary costs were noted as key determinants to participation in peer support programs. When asked about logistical concerns related to the research design, participants suggested that the frequency of the peer support program at twice a week might be too burdensome and that the preferred data collection methods for individuals with TBI might involve telephone administration (i.e., as opposed to self-completion online) and the use of short and simple questionnaires.

**Readiness and motivation to participate.** Participants described engagement in and continued participation in peer support programs as dependent on "readiness" and factors that influence this readiness. Participants described readiness to participate in peer support programs as being influenced by the individual's acceptance of his/her injury or stage of recovery and willingness to share his/her experiences or learn about others' experiences. Similarly, motivation was described as a key factor in the level of engagement in the program; partners may be motivated to participate in the program because of its potential benefits (e.g., learning new coping strategies); and mentors may be motivated to participate because of their desire to impart knowledge and advice to others with similar lived experiences. Participants shared that when they have a high sense of motivation, they prioritize the interaction and are more receptive to the program benefits. Capitalizing on one's readiness to participate and then appealing to his/her motivation to participate were perceived as key to recruitment and engagement in peer support programs. Parallel approaches were noted for facilitating recruitment, retention, and adherence in research.

**Need for clear expectations.** Clear expectations of what is involved in both the peer support research process and program were also identified as important considerations for

**Table 1. Significant quotes from themes related to barriers and facilitators to participating in peer support research and programs.**

| Theme | Quote | Source |
|---|---|---|
| Knowledge, awareness and communication | "We didn't hear about OBIA at the hospital. It was like over a year later we heard about it. So I think it's important maybe pamphlets, when they give out like those pamphlets and stuff like that in the hospitals." | Mentor—4 |
| | "Having some clear, simple instructions or information about why they're participating in the research and what the research aims to do and kind of what their role will be may be really helpful for them to have. And, I think, having something in front of them, as far as written communication, is sometimes a much better idea than providing information over the phone." | Peer support staff—2 |
| Logistics of participating | "And over the phone seems to work, because people can kind of stay in their own space where they're comfortable, and can probably have an easier time managing symptoms than, say, meeting face to face somewhere, where maybe there's really bright lighting or there are lots of busy visual things, which might be challenging for people." | Mentor and previous partner—7 |
| | "I would say the most important element is the fact that this program reaches people in their own home, in their own community. Especially, for individuals who don't have transportation into services or service providers, or are out in more rural populations, this allows them to participate in a program where they can get support and information, no matter where they're from. (. . .) The other, to me, is the fact that this program is free. There are not a lot of programs and services out there where people don't have to pay to participate in a program these days." | Peer support staff—2 |
| Readiness and motivation to participate | "But I think ultimately it is a certain level of, I don't know if it's acceptance as much as adjustment to the impact of their injury and that they feel emotionally prepared to share their story, hear one's story." | Peer support staff/ researcher—18 |
| | "I know that one of the things that I've seen that precludes a partnership being able to last is just that willingness to be involved. That has to be maintained because I signed up for this to help people and I want to try and do that but I didn't sign up for it to be a door to door salesman for the program. I'm not trying to convince them to take part in it all the time or that they should be very serious about it or anything else. I'd like to see that understood before the partnership begins." | Mentor—3 |
| Clear expectations | "One of the exercises that we did when we were in the mentorship training was to simulate a phone call and go through it. I think that might be beneficial on both sides. I mean, if we're getting somebody that's going to participate, they should actually indicate that they're with some understanding of how the phone call actually works because it's been difficult, it's been very frustrating." | Mentor—3 |
| | "So, when you're thinking about a randomised control trial, truly controlling the intervention will be your challenge because your intervention can go anywhere in peer mentorship. But for that reason, when you're doing that randomised control trial, assessing fidelity, or understanding the active ingredients of what happens in that conversation is going to become really important. Because even though you have the same peer again and again and again, there will be some cases where they do 25 reflections and provide all of this knowledge, and other cases where that bond doesn't happen and no knowledge is provided. And what you don't want to end up doing with a randomised control trial is saying that those two interactions are the same, and that for some reason peer mentorship doesn't work." | Researcher—13 |
| | "Well, I know that we're not supposed to exchange phone numbers. I, personally, think that is one of the better things that this lady and I have done. Because, it's kept us on track with speaking to each other once a week. I think, even though that's a rule, that's a rule that can be looked at again. I get they have it there so that the person that's the mentor is not always being phoned by the person they're partnering with. (. . .) We spoke about the priority of that at the very beginning, that she wouldn't be using that phone call for anything else other than to call me if she was not going to be available to talk to me at that time." | Mentor—10 |
| Matching *(theme for participating in peer support programs only)* | "I think it just depends on the connection that they make with whoever they're talking to. If they don't connect it's not going to work and this is why I wonder about the mentor that my son has. I don't know how well they connect." | Caregiver partner—8 |
| | "And interestingly, I think, from what we've learned with that, is that a key aspect of a good peer mentorship program is about ensuring that you have similar life experiences and that there is opportunity to share those experiences in a way that has a strong amount of emotional intelligence as well as a peer mentor that has a lot of knowledge and understanding of the topic. And the key piece on matching becomes about the experience and then less so some of those demographic characteristics that we would traditionally think make a great mentorship. So, not necessarily that I have the same level injury as you, but that maybe we both can't use our thumbs. And I'm trying to learn about that. But the other one was experience. Having similar experiences or interest in knowledge of similar experiences." | Researcher—13 |

ongoing commitment and adherence. Participants identified a need for clearly defined roles and responsibilities for both the mentors and partners. As mentors go through a training process, it is clearly outlined what is expected of them in the peer support process, and mentors indicated that setting up similar expectations for partners would also be beneficial. In terms of general expectations of peer support programs, while most peer support interactions were described as an organic process with topics of conversation occurring spontaneously, some participants indicated a more structured program guide may be beneficial. The unstructured nature of peer support interactions was identified as a challenge for conducting research, as peer support can be delivered in different ways, and therefore assessing the precise components which contribute to success might not be possible. Furthermore, the establishment of boundaries for the mentor and partner relationship was recognized as important to facilitate a more focused and professional interaction; however, it was noted that challenging these boundaries (e.g., exchanging personal phone numbers) also facilitated participation in the peer support program (i.e., more flexibility, able to reschedule calls directly rather than through a third party).

**Matching.**    Unsurprisingly, most participants indicated that compatible matches between mentors and partners gave rise to more positive experiences. Participants described "having similar experiences" (e.g., having injuries that may be different but affecting the same ability to do something or experiencing similar symptoms as a result of the injury) as one of the most important criteria for matching between mentors and partners with TBI. These criteria were cited as more important than traditional ideas of matching, such as age or gender. It was indicated that this matching would facilitate retention and adherence to the program specifically, but was not noted as a facilitator to the research process.

## Perceived impacts of peer support

All participants were asked about the perceived impact of participating in a peer support program; all three themes were derived inductively. Representative quotes of the perceived impacts of peer support have been compiled in Table 2.

**Acceptance, community, social experiences.**    This theme related to feelings of acceptance of having a brain injury or the reduction of stigma felt when connected with someone else with a similar experience. Participants indicated that a sense of community was fostered in sharing lived experiences. Peer support promoted opportunities to talk about brain injury related experiences, which were normalized or validated.

**Vicarious experience/learning through others: Shared experiences, role-modelling, encouragement.**    Participants noted that the sharing of lived experiences with a brain injury allowed mentors the opportunity to act as role-models for their partners and to discuss their strategies for coping with a brain injury (e.g., by staying positive and approaching tasks incrementally). Participants noted that through the process of sharing experiences, they were able to gain or impart knowledge as well as strategies and solutions related to issues of living every day with a brain injury. Participants indicated that encouragement was also used to help partners feel more confident to implement adaptations that would help them with their daily activities (e.g., cooking, knitting, and cleaning).

**"I feel better".**    The theme of "I feel better" captured participants' perceived sense of improved mood as a result of having a peer mentor. This sense of improvement was noticed by both the partners and the mentors themselves. While mood/severity of depression was recognized as an important outcome from an evaluation/research perspective, it was acknowledged that traditional measures for mood/depression might not capture the overall positive experience with the program or benefits one might accrue.

**Table 2. Significant quotes from the themes related to the perceived impacts of peer support.**

| Theme | Quote | Source |
|---|---|---|
| Acceptance, community, social experiences, and genuine friendships | *"I think one of the other aspects of this program is the connection between people who have lived experience, providing some support and some, maybe I could call it, normalcy to people who are newer to brain injury. (. . .) We know how vast and how varied brain injury is, and I think having somebody to help to normalize that experience can be really, really powerful."* | Peer support staff —2 |
| | *"It allows people to meet new people even if you are just talking on the phone. It's sort of increasing your circle if that makes sense. Your friends, of course, are always going to be there for you, but a lot of people don't understand brain injury and they don't get what makes people tick or not tick."* | Partner—17 |
| Vicarious experience/learning through others: shared experiences, role-modelling, encouragement | *"I find the conversations I have with my new partners are a lot more focused on brain injury and their symptoms. They often ask me about my experiences and I'm comfortable sharing that as well, so they'll ask about what my symptoms are like, how was returning to work, various concussions that I've had, and what that looks like and what I've found helpful for my recovery and that kind of thing.* | Caregiver mentor —22 |
| | *You share, for people you support, I guess you share some of the activities, and some of the ups and downs that you have gone through, and some of the solutions . . . and I've also been able to get some tips from people that I was supporting, just by chatting and seeing what they have done. I said, oh, that might be a good idea to try."* | |
| | *"She is really inspiring me or helping me to do the things that I did before but yet I stopped because I thought, because of the brain injury, that I wasn't able to understand. . . Or even, I can learn old things that I did know before yet I had lost all the confidence that I had before. . . Knitting, for example. I was an excellent knitter, but now, like, I had to understand and to relearn the stitches and that. So I ended up sewing or knitting dishcloths for a long time. It's boring, it's always the same thing, but she's also a knitter and she really encourages me to go ahead, and to try again."* | Partner—21 |
| "I feel better" | *"I think it definitely helps improve my mood, because I feel like there were times before I was connected with anyone that I was feeling pretty low and kind of despondent."* | Mentor and previous partner —7 |
| | *"One of the most apparent might be mood, because you have that subjective self-report of I feel better. And what does better mean? Do we want to slap a depression scale on them? I don't know. And I'm sure that may be involved in the study is a pre and post various mood-related questionnaires. But the challenge with that is, can you directly correlate the mood to that peer support intervention or the intangible? Tangible but yet intangible is the passage of time. And what are all these life events coming together to create that sense of acceptance? That will be very tricky to try and quantify in an RCT."* | Peer support staff —1 |

### Using integrated knowledge translation to design the pilot feasibility randomized controlled trial protocol

The themes resulting from the interviews were communicated to and discussed with OBIA to finalize the procedures for the pilot feasibility RCT. While some components of the proposed trial were validated, many components were modified. Three key changes were made as a result of the stakeholder interviews in combination with the consultation from OBIA:

1. The twice a week intervention arm, initially designed to measure the dose-response of peer support was judged to be too burdensome and was removed.

2. The length of trial was reduced from 6 months to 4 months. This change reflects findings from the interviews about the need to capitalize on readiness to participate, as well as expressed concerns from OBIA about hindering access to services at a time when individuals were ready and motivated to participate. In consultation with the literature, a 4-month waitlist appeared to be feasible for measuring change across time and was a more acceptable timeframe for OBIA.

3. The Community Integration Questionnaire (CIQ) replaced the Participation Assessment with Recombined Tools—Objective (PART-O) as an outcome measure. While the outcome of community integration/participation was validated in the interviews as a benefit of participating in peer support programs, a closer review of the proposed outcome measure with OBIA indicated that the PART-O (productivity, out and about, and social relations) may not capture the type of participation that is promoted via the program that OBIA provides as accurately as the CIQ (home integration, social integration, productive activities).

Furthermore, our stakeholder interviews identified the need for contextual considerations when working with individuals with TBI and discussions with OBIA led to a brainstorming of methods for addressing specific concerns. Issues such as memory, energy, and need for flexibility were cited as major considerations affecting all aspects of participation for individuals with brain injury. In response to this, procedures for reminder calls, check-ins, and flexibility in scheduling were implemented. Additionally, while we initially planned to implement online surveys, the importance of conducting surveys over the telephone (and not self-administered) became clear. We also consulted with OBIA to develop and modify recruitment materials that were clear and accessible (i.e., lay language). Table 3 outlines a summary of how the themes that emerged from the results and discussions with OBIA influenced changes to the pilot feasibility RCT.

## Discussion

Five main themes of the barriers and facilitators to participating in peer support research and programs, and three main themes of the perceived impacts of peer support were identified from interviews with key informants. Incorporating an iKT approach, these themes were communicated to OBIA, and together, we finalized the pilot feasibility RCT protocol. The themes obtained from the key informant interviews provided validation for the specific selection of the self-efficacy and mood outcome measures. Additionally, discussions with OBIA led to several significant adaptations to our trial protocol, including removing the twice/week intervention arm, shortening of the length of trial, and changing the measure for the community integration outcome.

Furthermore, the identification of positive impacts of peer support from the perspective of our key informants further highlighted peer support service as a key community resource of the healthcare system.

**Table 3. Changes made to the phase two trial RCT protocol when results were communicated to our research partner, OBIA.**

| Major Themes | Proposed Trial Protocol | Resultant Outcome of Discussion with Research Partner and Rationale |
|---|---|---|
| Knowledge, awareness and communication | Partner with provincial peer support coordinators to support the recruitment efforts. | A one-page, lay language summary of the pilot feasibility RCT protocol was created and we presented this study at the annual peer support coordinator meeting to ensure the provincial peer coordinators were aware of the study and its design, as well as their role in recruitment (S2 File); implemented regular reminders and check-ins about on-going recruitment. |
| Logistics of participating | Research arms to assess dose response: | Removed 2X/week intervention arm as it was identified as being too burdensome for partners |
| | 2X/week intervention (n = 20) | |
| | 1X/week intervention (n = 20) | |
| | Control group (n = 20) | |
| | Self-administration of outcome measures at baseline, 6 weeks, 3 months, and 6 months | Researchers to collect outcome measures with partners over the telephone at baseline, 2 months, and 4 months; implementation of reminder and follow-up calls |
| | Health Survey (SF-12) to measure health-related quality of life outcomes | Changed to Health Survey (SF-20) because of cost considerations; in review with our Research Partner, they felt it was not too much of an added burden to partners |
| Readiness and motivation to participate | Length of Trial: 6 months | Reduced length of trial to 4 months to capitalize on partners' readiness and motivation to participate; in particular, the 6 months waitlist length was judged to be too long. |
| Clear expectations | Partner with provincial peer support coordinators to support the recruitment efforts. | A one-page, lay language summary of the pilot feasibility RCT protocol was created and we presented this study at the annual peer support coordinator meeting to ensure the provincial peer coordinators were aware of the study and its design, as well as their role in recruitment (S2 File); implemented regular reminders and check-ins about on-going recruitment. |
| Acceptance, community, social experiences, and genuine friendships | PART-O* to measure community integration/participation outcomes | Changed to CIQ[†] to better capture the impact of peer support on the social aspect of community integration/participation |
| Vicarious experience/learning through others: shared experiences, role-modelling, encouragement | TBI Self-efficacy Questionnaire to measure self-efficacy outcomes | No change made; provided validation for selection of outcome measure |
| "I feel better" | PHQ-9[‡] to measure mood outcomes | No change made; provided validation for selection of outcome measure |

[*] Participation Assessment with Recombined Tools—Objective.

[†] Community Integration Questionnaire.

[‡] Patient Health Questionnaire.

## Participation in peer support research and programs, and the impact of peer support on individuals with traumatic brain injury

To the best of our knowledge, no studies have examined barriers and facilitators to participating in peer support research, and thus the current study provides insights into how to enhance peer support research in the TBI context, so that the benefits of peer support as a component of rehabilitation can be realized. For example, the current study identified readiness and motivation as key factors in participation in peer support research among individuals with TBI. Thus, there is a need for future peer support research and initiatives to further investigate the specific considerations that drive participation, so that sub-groups that tend not to participate in research and/or programs can be directly targeted. Although there is no previous literature on the barriers and facilitators to participating in peer support research, we have identified that the factors that are important to participating in peer support research are similar to those factors that are important to participating in peer support programs.

Previous literature on the barriers and facilitators to participating in peer support programs are consistent with the themes identified in this study. Our research team has previously

conducted a systematic review examining the impact of peer support interventions for individuals with acquired brain injury, cerebral palsy, and spina bifida. We identified that logistical challenges (e.g., geographical distance and scheduling difficulties) act as barriers to participation in peer support programs, but are mitigated by facilitators such as a common background and/or sense of identity between mentors and partners [12]. Furthermore, previous studies in other disease populations have identified barriers and facilitators to participation in peer support programs that overlap with many of the themes that emerged in the current study [35–37]. However, the need for clear expectations for both mentees (partners) and mentors, identified by our key informants as a major theme in this study, has not been previously reported. Furthermore, Taylor and colleagues identified that perceived stigma was a barrier to participating in peer support programs (i.e., patients' fear that healthcare providers would judge them as being incapable of managing their own treatment regimen) among individuals with chronic kidney disease [37]. Although we did not identify this specific barrier in our current study among individuals with TBI, a related issue was noted within the *knowledge, awareness, and communication* theme in that participation in peer support programs could be improved if healthcare providers initiated conversations about such programs with their patients.

Numerous positive perceived impacts of peer support in individuals with TBI were identified in the current study, consistent with benefits identified in previous studies. For example, improved feelings of acceptance and empowerment, knowledge of TBI, overall quality of life, ability to cope with depression, ability to obtain friendships, sharing of coping strategies, and perceived social support have been previously demonstrated [10–12, 38].

## Using integrated knowledge translation to inform a trial protocol

To the best of our knowledge, this is the first study to report using an iKT approach to inform a pilot feasibility RCT protocol, specifically the methods for data collection. Gagliardi and colleagues have identified that the involvement of key informants in research related activities has been poorly described, especially in data recruitment, collection and interpretation, and often consists of a one-way communication method where researchers simply provide key informants with research summaries, rather than engaging them in decision-making activities [24]. In contrast, in our study, we maintained regular communication with OBIA, early and final results were shared with OBIA, and the implications of the final results to the proposed trial protocol were discussed and agreed upon with OBIA (Table 3). Changes to the trial were made with the goal of increasing the *accessibility* (e.g., by avoiding the use of scientific jargon when communicating with key informants/partners), *relevance* (e.g., through careful selection of outcome measures), and *endurance* (e.g., by working in close partnership with OBIA) of the outcomes. In doing so, it is intended that peer support research will be more relevant and applicable to the needs of individuals with TBI, and therefore, enhance their participation in research and programs [39]. Thus, the unique contribution of this paper in part is an exemplar of the specific processes involved in iKT for the purposes of refining a trial protocol and directly linking these processes with meaningful outcomes (e.g., change in length of trial, production of lay language summary, etc.). Fidelity to the iKT approach will maximize its potential to produce research outcomes that are more applicable for healthcare practice and policy change, and should be promoted given the increasing application of iKT in health services research [24].

## Study limitations

The active recruitment of key informants likely led to selection bias. For example, the key informants who represented the TBI population were likely healthier (e.g., fewer cognitive

deficits), had positive experiences with peer support, and were more motivated to participate in our study, which may not be representative of the larger TBI population. We also included experts in KT and TBI who may not have had expertise in all content areas (i.e., TBI, peer support, RCTs), but were still able to provide varied and insightful perspectives given that they had expertise in at least one of the areas.

## Conclusions

The current study showcases peer support as a promising community intervention to aid in the rehabilitation of TBI survivors. The study also provides new insights into how to enhance peer support research and programs within the TBI population. Furthermore, it provides an exemplar of the specific processes involved in iKT for the purposes of refining a pilot feasibility RCT protocol and linking these processes to outcomes that are meaningful to key stakeholders.

## Supporting information

**S1 File.**
(PDF)

**S2 File.**
(PDF)

**S1 COREQ checklist.**
(PDF)

## Author Contributions

**Conceptualization:** Dorothy Luong, Shane N. Sweet, Mark Bayley, Ben B. Levy, Monika Kastner, Michelle L. A. Nelson, Nancy M. Salbach, Susan B. Jaglal, John Shepherd, Ruth Wilcock, Carla Thoms, Sarah E. P. Munce.

**Data curation:** Dorothy Luong, Sarah E. P. Munce.

**Formal analysis:** Stephanie K. C. Lau, Dorothy Luong, Shane N. Sweet, Mark Bayley, Ben B. Levy, Monika Kastner, Michelle L. A. Nelson, Nancy M. Salbach, Susan B. Jaglal, John Shepherd, Ruth Wilcock, Carla Thoms, Sarah E. P. Munce.

**Funding acquisition:** Sarah E. P. Munce.

**Investigation:** Dorothy Luong, Sarah E. P. Munce.

**Methodology:** Dorothy Luong, Mark Bayley, Sarah E. P. Munce.

**Project administration:** Stephanie K. C. Lau.

**Supervision:** Sarah E. P. Munce.

**Writing – original draft:** Stephanie K. C. Lau, Dorothy Luong, Sarah E. P. Munce.

**Writing – review & editing:** Stephanie K. C. Lau, Dorothy Luong, Shane N. Sweet, Mark Bayley, Ben B. Levy, Monika Kastner, Michelle L. A. Nelson, Nancy M. Salbach, Susan B. Jaglal, John Shepherd, Ruth Wilcock, Carla Thoms, Sarah E. P. Munce.

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
