## [Decision Letter · Decision Letter 0]

16 Jun 2021

PONE-D-21-06700

Using an integrated knowledge translation approach to inform a pilot feasibility randomized controlled trial on peer support for individuals with traumatic brain injury: a qualitative descriptive study

PLOS ONE

Dear Dr. Lau,

Thank you for submitting your manuscript to PLOS ONE. After careful consideration, we feel that it has merit but does not fully meet PLOS ONE’s publication criteria as it currently stands. Therefore, we invite you to submit a revised version of the manuscript that addresses the points raised during the review process.

We look forward to receiving your revised manuscript.

Kind regards,

Alfio Spina, M.D.

Academic Editor

PLOS ONE

Journal Requirements:

2.Please provide additional details regarding participant consent. In the ethics statement in the Methods and online submission information, please ensure that you have specified what type you obtained (for instance, written or verbal, and if verbal, how it was documented and witnessed). If your study included minors, state whether you obtained consent from parents or guardians. If the need for consent was waived by the ethics committee, please include this information.

3. Please note that according to PLOS ONE policies,  if materials, methods, and protocols are well established, authors may cite articles where those protocols are described in detail, but the submission should include sufficient information to be understood independent of these references (https://journals.plos.org/plosone/s/submission-guidelines#loc-materials-and-methods).Thus, we would recommend that the methods used for  sampling and recruitment of participants, and for data analysis, are reported in more detail in the Methods section.

We will update your Data Availability statement to reflect the information you provide in your cover letter."

Additional Editor Comments (if provided):

Reviewer #1: The aim of this article is two explore key informants’perspectives (iKT) of barriers and facilitators of participating in peer support interventions and of its effectiveness, among people with TBI. Finally, the authors try to demonstrate how iKT can be useful in the development of a RCT.

The article is well-written in a correct English.

TBI affects millions of people each year worldwide. In addition to the physical and cognitive deficits, TBI can lead to depression, anxiety and reduced quality of life. A promising and cost-effective interventions is peer support.

The topic of the article is interesting and not extensively development yet in neurosurgery. However, the article is, currently, too long and not completely focused on the aim of the study. The article is interesting but it is too long, this means that it is difficult to keep the attention throughout the reading. I think that the paper should be shorter and more tailored on the aim of the study.

Reviewer #2: Integrating peer support into rehabilitation of severe TBI‘S patients is an interesting topic. However, multicenter studies are needed to validate utility in different countries.

Furthermore this study shows the useful integration of iKT in the construction of the RCT protocol.

the work is well written and I thank the authors for their submission.

Academic Editor: Please revise the article according the Reviewers' comments. Reduce the lenght of the manuscript to improve its readability.
---

## [Author Response · Author response to Decision Letter 0]

8 Jul 2021

Dear Editor:

Thank you for your email. I am pleased to hear that you believe that our manuscript will have the potential to be published in PLOS ONE following requested revisions.

As per your request, please find our response to the peer-reviewers’ comments with every change outlined point by point below:

Editor and Reviewer comments: 

Reviewer #1: "The aim of this article is two explore key informants’perspectives (iKT) of barriers and facilitators of participating in peer support interventions and of its effectiveness, among people with TBI. Finally, the authors try to demonstrate how iKT can be useful in the development of a RCT.

The article is well-written in a correct English.

TBI affects millions of people each year worldwide. In addition to the physical and cognitive deficits, TBI can lead to depression, anxiety and reduced quality of life. A promising and cost-effective interventions is peer support.

The topic of the article is interesting and not extensively development yet in neurosurgery. However, the article is, currently, too long and not completely focused on the aim of the study. The article is interesting but it is too long, this means that it is difficult to keep the attention throughout the reading. I think that the paper should be shorter and more tailored on the aim of the study."

 - We thank the reviewer for taking the time to provide their feedback on our manuscript. We have taken this feedback into consideration when shortening our manuscript to being more focused on the objectives. 

Reviewer #2: "Integrating peer support into rehabilitation of severe TBI‘S patients is an interesting topic. However, multicenter studies are needed to validate utility in different countries.

Furthermore this study shows the useful integration of iKT in the construction of the RCT protocol.

the work is well written and I thank the authors for their submission."

 - We thank the reviewer for taking the time to provide their feedback on our manuscript.

Academic Editor: "Please revise the article according the Reviewers' comments. Reduce the lenght of the manuscript to improve its readability."

 - We thank the editor for taking the time to provide their feedback on our manuscript. We have reduced the length of the manuscript as suggested. 

• Again, we thank the reviewers and editor for providing their feedback on our manuscript and we have taken all their feedback into consideration when revising our manuscript. Additionally, as per the journal requirements, our manuscript meets PLOS ONE’s style requirements, we specified the participant consent process, we elaborated on our methods section, and we reviewed our reference list to ensure both completeness and correctness. However, in regard to the minimal underlying data set, the relevant data are within the manuscript/supporting information files, and we cannot share the full transcripts publicly as the participants did not consent to such. However, researchers can request for access to these transcripts via the University Health Network’s Research Ethics Board. 

Should your editorial office require any further edits following my most recently submitted submission, please do not hesitate to inform me and we will make these changes as soon as possible. Thank you for your consideration.

Yours sincerely,

Stephanie KC Lau, BSc, MD Candidate, on behalf of the study authors

SLau065@uottawa.ca

---

## [Editor Report · Decision Letter 1]

12 Aug 2021

Using an integrated knowledge translation approach to inform a pilot feasibility randomized controlled trial on peer support for individuals with traumatic brain injury: a qualitative descriptive study

PONE-D-21-06700R1

Dear Dr. Lau,

We’re pleased to inform you that your manuscript has been judged scientifically suitable for publication and will be formally accepted for publication once it meets all outstanding technical requirements.

Kind regards,

Alfio Spina, M.D.

Academic Editor

PLOS ONE
---

## [Editor Report · Acceptance letter]

16 Aug 2021

PONE-D-21-06700R1 

Using an integrated knowledge translation approach to inform a pilot feasibility randomized controlled trial on peer support for individuals with traumatic brain injury: a qualitative descriptive study 

Dear Dr. Lau:

I'm pleased to inform you that your manuscript has been deemed suitable for publication in PLOS ONE. Congratulations! Your manuscript is now with our production department. 

Kind regards, 

on behalf of

Dr. Alfio Spina 

Academic Editor

PLOS ONE